biomaterials/bioengineering/biomimetics

bone cement, biocompatibility, hydroxyapatite, poly(ε-caprolactone), poly(methyl methacrylate), surface grafting

**Authors for correspondence:**
I-Ming Chu
e-mail: imchu@che.nthu.edu.tw
Po-Liang Lai
e-mail: polianglai@gmail.com

†These authors contributed equally to this study.

This article has been edited by the Royal Society of Chemistry, including the commissioning, peer review process and editorial aspects up to the point of acceptance.

# Incorporation of surface-modified hydroxyapatite into poly(methyl methacrylate) to improve biological activity and bone ingrowth

Kuan-Lin Ku[1,†], Yu-Shan Wu[1,†], Chi-Yun Wang[2,5],
Ding-Wei Hong[1], Zong-Xing Chen[2,5], Ching-An Huang[3],
I-Ming Chu[1] and Po-Liang Lai[2,4,5]

[1]Department of Chemical Engineering, National Tsing Hua University, No. 101, Section 2, Kuang-Fu Road, Hsinchu 30013, Taiwan, Republic of China
[2]Department of Orthopedic Surgery, Chang Gung Memorial Hospital, No. 5, Fuxing Street, Guishan District, Taoyuan City 33305, Taiwan, Republic of China
[3]Department of Mechanical Engineering, Chang Gung University, Taoyuan, Taiwan, Republic of China
[4]College of Medicine, Chang Gung University, Taoyuan, Taiwan, Republic of China
[5]Bone and Joint Research Center, Chang Gung Memorial Hospital, Taoyuan, Taiwan, Republic of China

P-LL, 0000-0002-2020-919X

Poly(methyl methacrylate) (PMMA) is the most frequently used bone void filler in orthopedic surgery. However, the interface between the PMMA-based cement and adjacent bone tissue is typically weak as PMMA bone cement is inherently bioinert and not ideal for bone ingrowth. The present study aims to improve the affinity between the polymer and ceramic interphases. By surface modifying nano-sized hydroxyapatite (nHAP) with ethylene glycol and poly(ε-caprolactone) (PCL) sequentially via a two-step ring opening reaction, affinity was improved between the polymer and ceramic interphases of PCL-grafted ethylene glycol-HAP (gHAP) in PMMA. Due to better affinity, the compressive strength of gHAP/PMMA was significantly enhanced compared with nHAP/PMMA. Furthermore, PMMA with 20 wt.% gHAP promoted pre-osteoblast cell proliferation *in vitro* and showed the best osteogenic activity between the composites tested *in vivo*. Taken together, gHAP/PMMA not only improves the interfacial adhesion between the nanoparticles and cement, but also increases the biological activity and affinity between the osteoblast cells and PMMA composite cement. These results show that gHAP and its use in polymer/bioceramic composite has great potential to improve the functionality of PMMA cement.

# 1. Introduction

Poly(methyl methacrylate) (PMMA) is well known as bone cement due to its wide utilization in orthopedic surgery in procedures such as osteoporotic vertebral fractures, treatment of bony metastases and fixation of components in arthroplasties [1–4]. PMMA bone cement provides not only immediate pain relief, but also mechanical stability [5]. PMMA is an acrylic polymer and is formed by the mixture of a liquid MMA monomer and a powder MMA polymer. It has been the only material used for anchoring artificial replacements to contiguous bones [6]. However, the physical and chemical properties of PMMA are far from ideal from a spine surgeon's point of view. Poor biological activity of PMMA leads to poor interaction between the local bone and PMMA cement [4,6].

Several kinds of ceramics, such as $TiO_2$ nanoparticles [7,8], tricalcium phosphate (TCP) [9], brushite and nHAP [10], have been incorporated into PMMA cement to enhance biological activity of PMMA cement. Nano-sized hydroxyapatite (nHAP) is a widely used bioceramic material for bone regeneration due to its similarity to natural bone components and its good biological activity [11]. nHAP also has excellent osteoconductivity and good maintenance of mechanical properties [12]. Additionally, the incorporation of nHAP into PMMA cement increases the viscosity of cement and thus facilitates cement injectability during the surgical process [13]. However, nHAP has poor capability of dispersion in organic solvents and tends to agglomerate in a polymer matrix when added by blending [14,15]. The poor dispersion of nHAP not only generates agglomeration, but also hinders the functionality of nHAP [15,16].

To date, nHAP can be modified with polymers via hydroxyl groups on its surface. It has been reported that the affinity between surface-modified nHAP and the polymer matrix could be improved and enhance the colloidal stability of the particles in the organic solvents [17,18]. Due to the difficulties of initial grafting polymerization on the surface of nHAP, there is a low grafting efficiency on the surface with less than 10 wt.% [17,19]. Lee *et al.* [17] introduced an approach using a two-step modification with a high polymer grafting efficiency. nHAP was first grafted with ester monomer to generate more functional groups and was subsequently modified with another type of polymer to achieve a high grafting efficiency [15].

Poly(ε-caprolactone) (PCL), which is one of the most promising bioresorbable polymers and is widely investigated as a scaffold in bone tissue engineering owing to its non-toxicity and biocompatibility [20–22]. In this study, we have used PCL to surface modify nHAP. We hypothesized that a surface-modified nHAP on a PMMA-based composite enhances the dispersion of particles in a polymer matrix and improves the functionality of nHAP. A thermogravimetric analyzer (TGA) and a transmission electron spectrometer (TEM) were used to characterize the efficiency of the surface modification on nHAP. The biological activity of PMMA-based composites and bone ingrowth were demonstrated in both *in vitro* and *in vivo* studies.

# 2. Material and methods

## 2.1. Modification of nHAP

Nano-sized (less than 200 nm) HAP (nHAP) particles (Sigma-Aldrich, USA) were modified using a two-step ring opening reaction. nHAP particles were dried in the vacuum oven at 100°C 3 days prior to use. To create more hydroxyl groups on nHAP particles, the particles were first grafted with ethylene glycol (EG; J.T. Baker, USA). Then, 3 g nHAP particles were dispersed into 60 ml dry *N*, *N*-dimethylmethanamide (DMF, Tedia, USA) and modified with 5.86 g EG at 60°C overnight. These particles were then washed by dichloromethane (DCM, Macron Fine Chemicals, USA) and centrifuged at 3000 r.p.m. three times for 30 min each time. After being dried, 2 g of EG-modified nHAP (E-HAP) was suspended in 50 ml of dry toluene (ECHO chemical, Taiwan) and was reacted with 5 ml ε-caprolactone (ε-CL, Alfa Aesar, USA) at 130°C for 24 h. Free polymers were washed off using DCM. Then, 100 ppm of stannous octoate (Sn(Oct)2, Sigma-Aldrich) was used in these two reactions as a catalyst. PCL-grafted E-HAP (gHAP) was collected and dried in a vacuum oven.

## 2.2. Characterization of E-HAP and gHAP

Grafting ratios of the E-HAP and gHAP were characterized by thermogravimetric analysis (TGA, SSC5000, Seiko) at a heating rate of 5°C min$^{-1}$ from room temperature to 700°C under nitrogen gas

flow. The morphologies of nHAP and gHAP were observed by transmission electron spectrometer (TEM, JEM-2100, Jeol) where the particles were dispersed in DCM (at $1 \, \mathrm{g \, l^{-1}}$) and then dropped onto the TEM cupper grid with a carbon coating. The foils were air-dried at room temperature before observation.

## 2.3. Preparation of nHAP/PMMA and gHAP/PMMA composites

The composites were made from polymethylmethacrylate (Simplex® P PMMA; Stryker Howmedica, USA) with nHAP or gHAP particles at different ratios. The PMMA composite blended with 10 wt.% and 20 wt.% nHAP was named PMMAn10 and PMMAn20, respectively, while the PMMA composite blended with gHAP was named PMMAg10 and PMMAg20, respectively. PMMA without HAP particles was used as the control. The mixed ratio of the monomer liquid to the polymer powder was $0.5 \, \mathrm{ml \, g^{-1}}$. Then, the mixture was shaped in a silica mould for curing into tablets that had a diameter of 18 mm and cylinders with a diameter of 3 mm and a height of 12 mm.

## 2.4. Characterization of nHAP/PMMA and gHAP/PMMA composites

The morphologies of composite surfaces were observed by field emission scanning electron microscopy (FE-SEM, SU8010, Hitachi). The composition analysis of composite surfaces was determined by energy dispersive spectrometer system (EDX, Horiba, Hitachi).

The compressive strength of the cylindrical samples was measured using a universal testing machine (Bionix 858, MTS) at a compressing speed of $1 \, \mathrm{mm \, min^{-1}}$. The data were recorded at 0.5 Hz with the end level of $-2$ mm. For each group, five samples were tested, and the mean values were calculated.

## 2.5. *In vitro* assay for cell viability and cell attachment

A live/dead fluorescence viability assay kit was used to distinguish dead cells (red) from live cells (green). After cell culturing ($10^4$ cells/tablet) for 1 day, the culture medium was removed. The tablets were rinsed twice with PBS. Then, 1 ml of live/dead staining working solution (volume ratio, PBS: calcein AM: ethidium homodimer $= 1000 : 2 : 1$) was added into each well. After incubation at 37°C for 15 min, the live and dead cells were observed under confocal microscopy (TCS, SP8X, Leica).

Cell Counting Kit-8 (CCK-8) (Enzo Life Sciences, USA) was used to determine the viable cell attached on the surface of the tablets. MC3T3-E1 cell line is a pre-osteoblast cell line from mouse calvaria [23]. The tablets were placed into a 12-well plate, and then MC3T3-E1 cells were seeded ($10^4$ cells/tablet) on the surface of tablets ($n = 4$) for 1 day. Subsequently, the culture medium was removed, and 50 µl CCK-8 reagent and 450 µl α-MEM (Gibco, UK) were added into each well for 4 h incubation at 37°C. In the viable cells, highly water-soluble tetrozolium salt in CCK-8 reagent was reduced by active dehydrogenase to the water-soluble yellow-colour formazan. Thus, the amount of the yellow-colour formazan was proportional to the number of viable cells in a culture medium. After incubation, the CCK-8-containing supernatant was moved to the other new 96-well plates without tablet. A microplate reader (UV–Vis 8500, Tianmei) was used to measure the absorbance at 450 nm.

## 2.6. Cell morphology and cell proliferation on the surface of the composite

To observe the cell morphology and cell proliferation on the tablet surfaces, MC3T3-E1 ($5 \times 10^3$ cells) were seeded on the surface of each tablet per well in the 12-well culture plate, and incubated at 37°C for 3 days. The medium was changed every 12 days. After incubation for 3 days, the cell-seeded composite tablets were washed twice with PBS and then immersed into a 2% glutaraldehyde solution at 4°C overnight. The samples were then washed three times with deionized water and subsequently dehydrated by immersing into gradient solutions with increasing ethanol concentrations (50, 70, 95 and 100%) for 10 min in each solution. The tablets were vacuum dried and coated using platinum–palladium sputtering. SEM was used to observe the cell morphology on the tablet (JSM5600, Jeol).

## 2.7. Alkaline phosphatase assay

The alkaline phosphatase (ALP) assay kit (Abcam, USA) was used to evaluate the differentiation of the pre-osteoblastic cells MC3T3-E1, after these cells attach and proliferate on the surfaces of nHAP/gHAP with PMMA composites. In short, $2.5 \times 10^3$ MC3T3-E1 cells were seeded on the surface of the tablets and were cultured in osteogenic medium for 1, 7, 14 and 21 days. The osteogenic induction medium

is α-MEM (GIBCO, USA) containing 10% FBS (Corning Life Sciences, USA), 1% penicillin/streptomycin (Corning Life Sciences, USA), 0.1 µM dexamethasone (Sigma-Aldrich), 10 mM β-glycerol phosphate (Sigma-Aldrich) and 50 µg ml$^{-1}$ ascorbic acid (Sigma-Aldrich). Cell lysates were harvested and a bicinchoninic acid protein assay kit (Thermo Scientific, USA) was used for quantification of protein. Further, the ALP activity was normalized to total protein concentration. A microplate reader (UV–Vis 8500, Tianmei) was used to measure the absorbance at 450 nm.

## 2.8. In vivo animal study

Six male New Zealand white rabbits with weights from 2.5 to 3.0 kg (offered by Animal Health Research Institute, Taiwan) were used for this experiment. The surgical procedures were approved by the Animal Intuitional Review Board of Chung Gung Memorial Hospital and were conducted in compliance with the regulations for the care of laboratory animals. Here, we used a critical size defect rabbit model [24]. Briefly, each rabbit was anesthetized with an intramuscular injection of Zoletil 50 (10–15 mg kg$^{-1}$ body weight) and xylazine hydrochloride (5–10 mg kg$^{-1}$ body weight). The femurs of each rabbit were clipped and scrubbed. A 3 cm longitudinal skin incision was made in the lateral femoral condyle of the rabbit. Then, the femoral condyle was exposed. An electric drill created a critical size defect with a diameter of 4 mm in the centre of the lateral femoral condyle. The cements with volumes of approximately 100 µl were implanted into the bone defect site.

After three months, femurs with PMMA cements were fixed in 10% neutral formalin solution for one week before further processing. The morphology was observed using micro CT (NanoSPECT/CT, Mediso). The proximal epiphysis was cut off and fixed with an acrylic mounting kit (APISC). The specimen was sectioned using a diamond knife (Isomet, Buehler) with a thickness of 1 mm. The sectioned specimen was attached to a glass slide with an adhesive kit (M-Bond 610, VPG) and was then milled to a thickness of approximately 80 µm using 800-grit and 2000-grit sandpapers. The specimens with the target thickness were then processed for hemotoxylin & eosin (H&E) staining.

## 2.9. Statistical analysis

Statistical analysis and the results are shown as the mean $\pm$ s.d. Significant differences were evaluated by using unpaired two-tailed Student's $t$-test. The level of statistical significance was set at $p < 0.05$. *$p < 0.05$; **$p < 0.01$; ***$p < 0.001$.

# 3. Results and discussion

## 3.1. Efficient modification of HAP with PCL by two-stage method

To improve the affinity between the polymer and ceramic interphases, we modified nHAP surface with polymer by two-stage method. The grafting ratio of the polymers on the surface of nHAP and the changes in the morphology are shown in figure 1. TGA was used to detect the grafting ratio of particles. The grafting ratio was calculated with the weight differences of the non-grafted and grafted particles at 450°C after the polymers were completely burnt off without further weight loss. As shown in figure 1a, the TGA result of particles without or with polymer modification demonstrates that the grafting ratio was estimated to be approximately 14 wt.% for E-HAP and approximately 22 wt.% for gHAP. This result indicates grafting by two-stage method is more efficient. Then, TEM micrographs of particles without or with PCL modification (figure 1b) shows that the particles before and after the grafting procedures both maintained a spherical morphology, and a corona was clearly seen around gHAP, which is an indication of successful grafting of polymers to nHAP particles.

Further, we used SEM to examine the surface morphology of the composites tablet and results are shown in figure 2. Some bright spots were observed on the surface of PMMAn10 and PMMAn20 tablets (figure 2a,b). Elemental analysis was performed using an EDX detector on the bright area on the surface of PMMAn20, which is marked with a red cross in figure 2b. The bright spots on the surface of PMMAn20 were confirmed for the composition containing calcium and phosphate using EDX analysis. A closer look at the area revealed an agglomeration of particles in the PMMA matrix, as shown in the right top corner of the EDX spectrum (figure 2c). Conversely, there were no bright spots observed on the surfaces of PMMAg10 and PMMAg20 (figure 2d,e), which indicates that no significant particle agglomeration occurred in these two tablets. Plain PMMA surfaces were also

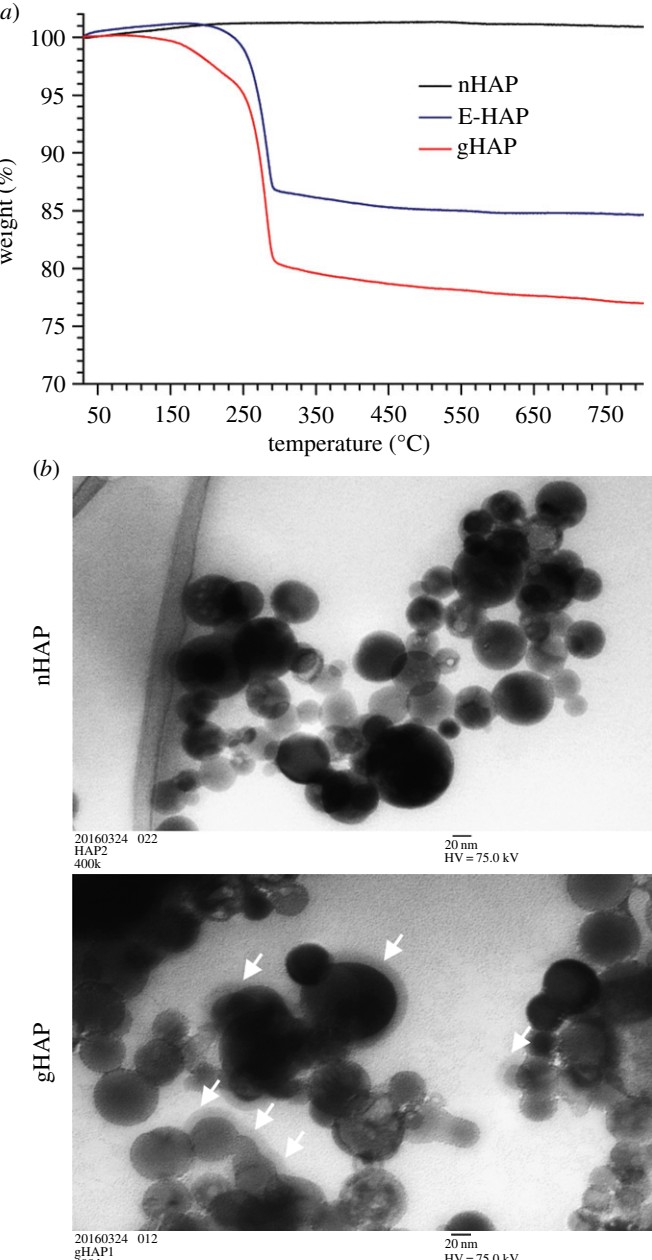

**Figure 1.** Grafting ratio and morphology change before and after surface polymer modification. (*a*) TGA results of HAP without grafting (nHAP) in black, ethylene glycol-grafted HAP (E-HAP) in blue and PCL-grafted E-HAP (gHAP) in red. (*b*) TEM images of nHAP and gHAP. White arrows point out the corona ring.

presented for comparison and had smooth surfaces with some small dots. The dots were attributed to some defects generated during tablet fabrication. Taken together, these results suggest that gHAP has better dispersion properties than nHAP.

Owing to interparticulate van der Waals forces and hydrogen bonding between the surface hydroxyl groups, nHAP particles tended to agglomerate and form clusters [16,25] under non-solvent condition. From previous studies, nHAP had worse colloidal stability and precipitated within 1 min in organic solvents [26]; however, the surface-modified gHAP with polymers had good colloidal stability [17,27]. The colloidal stability of nHAP particles affects the dispersion of these particles in the polymer matrix [17,28], which is corroborated in this study.

## 3.2. Compressive strength of nHAP (gHAP)/PMMA composites

The compressive strength of PMMA composite substitutes was determined by the failure point with an end level of −2 mm, and the results are shown in figure 3. The compressive strengths of PMMA,

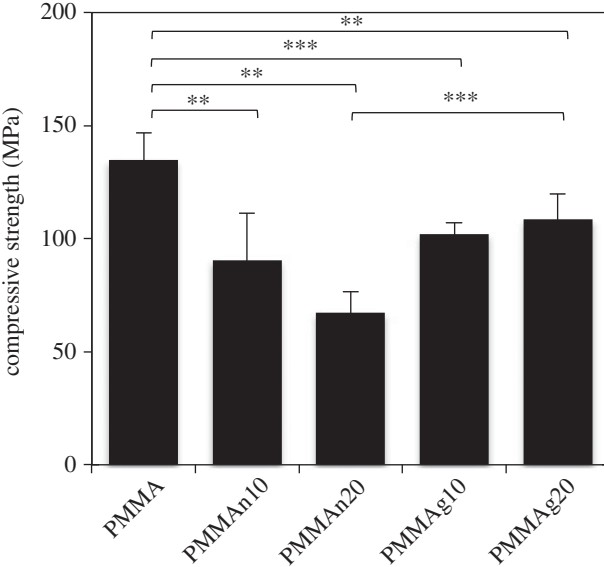

**Figure 2.** SEM images and EDX analysis of PMMA composite. SEM micrographs of PMMAn10 (*a*) and PMMAn20 (*b*) show some bright areas on the surface which were identified as the agglomeration of nHAP particles by EDX (*c*). PMMAg10 (*d*) and PMMAg20 (*e*) had a smooth and homogeneous surface similar to PMMA (*f*).

**Figure 3.** Compressive strength of PMMA-based composites. The mechanical property of the PMMA composite is tested with different percentages of nHAP and gHAP. The compressive strength decreased with an increase of nHAP loading. However, gHAP/PMMA maintained its compressive strength with an increase of gHAP loading. Five samples were tested, and the mean values were calculated. \*\**p* < 0.01 and \*\*\**p* < 0.001.

PMMAn10, PMMAn20, PMMAg10 and PMMAg20 were $134.8 \pm 11.9$, $90.5 \pm 20.6$, $67.3 \pm 9.3$, $103.8 \pm 3.3$ and $101.9 \pm 15.2$ MPa, respectively. The compressive strengths decreased significantly with an increase in the loading of the HAP particles in PMMAn10, PMMAn20, PMMAg10 and PMMAg20. These results indicate that gHAP/PMMA composites have better mechanical properties than nHAP/PMMA composites.

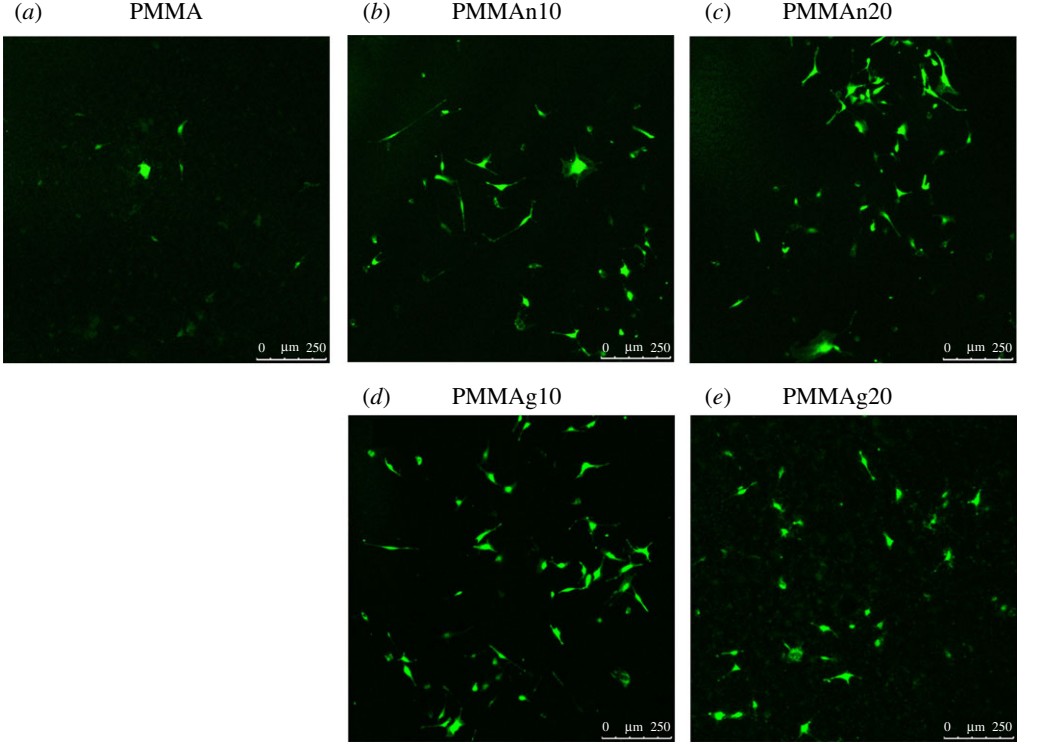

**Figure 4.** Live/dead fluorescence images of MT3T3-E1 cells ($1 \times 10^4$) seeded on PMMA (*a*), PMMAn10 (*b*), PMMAn20 (*c*), PMMAg10 (*d*) and PMMAg20 (*e*) for 1 day. There were more live cells on the surface of PMMA-based composites with an increased loading of nHAP and gHAP.

Previous studies [15,29] indicated that the mechanical properties decreased along with the increase of the loading of nHAP particles. The incorporation of nHAP particles into the polymer matrix prevents spread and movement of polymers within the composite [29]. PCL chains grafted on the surface of nHAP particles may generate more entanglements with polymer matrix due to the polymer–polymer chain interaction, and thus facilitate the interface adhesion between HAP fillers and a polymer matrix [30]. The mechanical properties could be improved by incorporating surface-modified particles into the composite materials due to better dispersion of the particles and good affinity between ceramic particles and polymers [15,17]. Poor adhesion between the components causes a decrease in yield stress as if the system is filled with voids. The compressive strength of PMMAg20 was significantly enhanced compared to PMMAn20 ($p < 0.01$; figure 3). It can be inferred that PMMAg20 had better dispersion of gHAP particles in PMMA cement and a favourable interface adhesion between gHAP and PMMA. Although the addition of gHAP to PMMA cement caused the decrease of overall mechanical properties, gHAP/PMMA composites still maintain a certain mechanical strength and are sufficient to fit most biomedical applications. The uniform dispersion of gHAP particles within PMMA cement and favourable interfacial adhesion between PMMA and gHAP may have a positive influence on the mechanical durability after longer exposure to body environment.

## 3.3. Viable cell attached on the composite tablets

To observe the viable cell attachment on the surface of composites, we seeded MC3T3-E1 cells on the nHAP/PMMA or gHAP/PMMA composites for just 1 day following live/dead staining. By live/dead staining, live cells are shown as green and dead cells are shown as red. In our results, there are almost no dead cells shown on the tablets. Data show that more live cells attached on the surface of tablets with a higher percentage of either nHAP or gHAP compared to PMMA only (figure 4). The results also point out a slightly better cell attachment in the gHAP/PMMA group (figure 4*d*,*e*) than in nHAP/PMMA (figure 4*b*,*c*). In addition, CCK-8 assay was used to quantitate the viable cell attachment on plain PMMA, nHAP/PMMA and gHAP/PMMA composites. In figure 5, the results show that an increased proportion of HAP particles increases cell viability. From this result, gHAP/ PMMA extracts (PMMAg10 and PMMAg20) have better cell viability than nHAP/PMMA composites

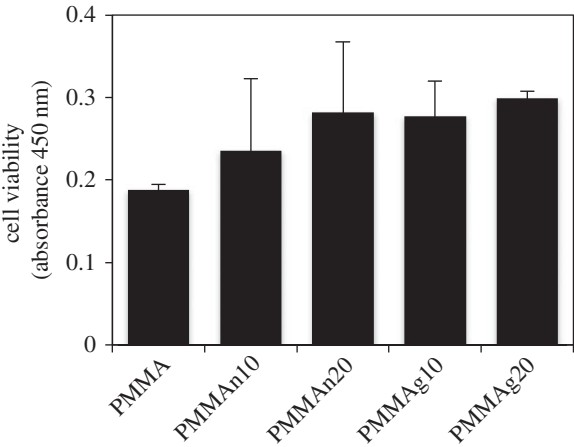

**Figure 5.** Cell viability of MT3T3-E1 cells on PMMA-based composites. $1 \times 10^4$ MT3T3-E1 cells were seeded in each well on 12-well culture plate as control or on tablets. The tablets are PMMA only or PMMA-based composites blended with 10 wt.% and 20 wt.% nHAP or gHAP. After incubation for 1 day, CCK-8 was used to evaluate the cell viability. Error bars show standard deviation.

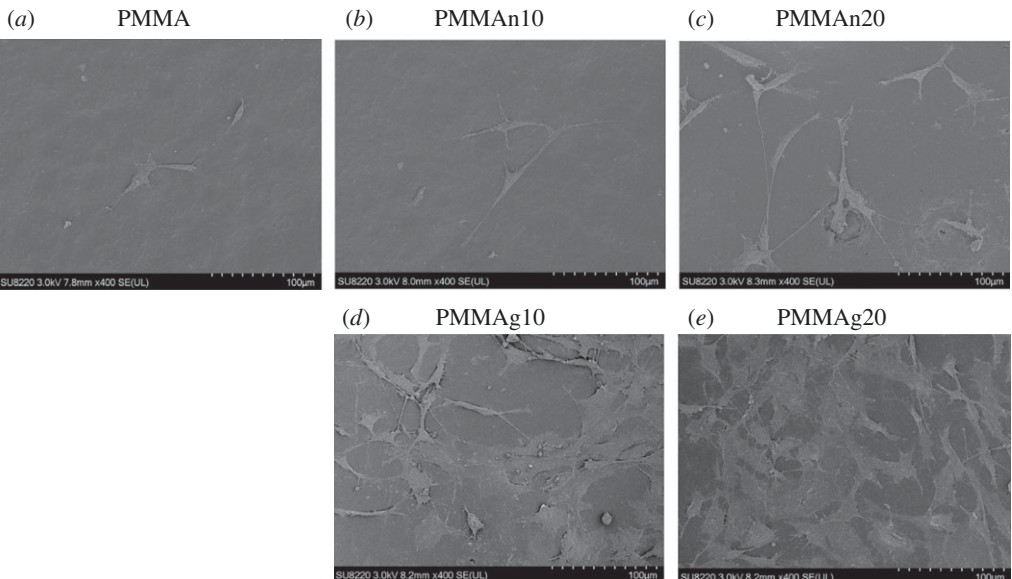

**Figure 6.** SEM images of cells seeded on tablets PMMA (*a*), PMMAn10 (*b*), PMMAn20 (*c*), PMMAg10 (*d*) and PMMAg20 (*e*) surfaces and then incubate for 3 days.

(PMMAn10 and PMMAn20). In addition, better cell distribution on gHAP/PMMA composites (PMMAg10 and PMMAg20) than on the nHAP/PMMA composites (PMMAn10 and PMMAn20) was observed from the live/dead fluorescence images. This result could be related to the better dispersion of gHAP in PMMA cement.

## 3.4. Morphology of cells and cell proliferation on the composite tablets

The cell morphology on the surface of the PMMA-based composite tablets was observed by SEM and was shown in figure 6. More cells were on the surface of tablets with an increase in the proportion of nHAP particles in PMMA, PMMAn10 and PMMAn20 (figure 6*a*–*c*). Additionally, compared to the surface of nHAP/PMMA, more cells were seen on the surface of gHAP/PMMA, especially PMMAg20 (figure 6*d,e*). In addition, on the surfaces of nHAP/PMMA composites (PMMAn10 and PMMAn20), cell clumps could be found; however, on the surfaces of gHAP/PMMA composites (PMMAg10 and PMMAg20), cells appeared to be distributed more homogeneously. In addition to the better cell morphology on the gHAP, after 3-day incubation, cell proliferation on the tablet is significantly

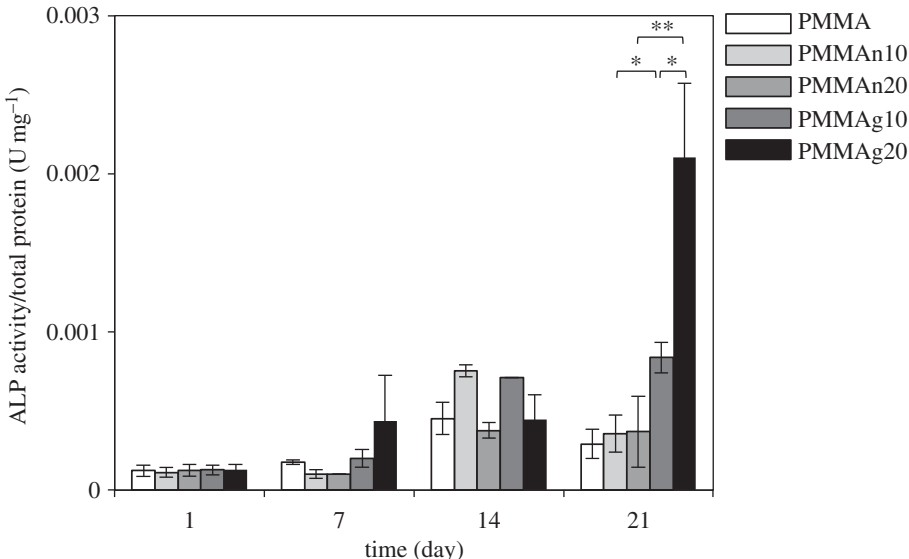

**Figure 7.** ALP activity of cells on composites. ALP activity increased with time. PMMAg10 and PMMAg20 showed greater ALP activity than PMMAn10 and PMMAn20 on day 21. Error bars show standard deviation. $*p < 0.05$, $**p < 0.01$.

increased. Taken together, an increased number of gHAP particles and their more uniform distribution in the composite tablets causes better cell attachment and cell proliferation.

HAP is a widely used biomaterial with good biocompatibility and cell affinity [12]. From the SEM results, the dispersion of HAP affected the cell distribution on the surface of the composite tablet, and this is also observed from the live/dead staining images. The agglomeration of nHAP in PMMA cement may create an obstacle for the nHAP–cell interaction, which diminished the function of nHAP in the composite tablets. The dispersion of gHAP in PMMA was better than nHAP, so cell attachment and cell proliferation on the composite tablet can be improved by the gHAP particles.

Kim et al. [31] have reported that cell proliferation of mouse fibroblast is significantly enhanced on 24 wt.% PCL-grafted HAP nano-composites compared to the PCL only because of excellent dispersion. Keivani et al. [32] have demonstrated that even the addition of 7 wt.% PCL-grafted HAP nanoparticles to PCL is enough to significantly increase cell proliferation of fibroblast. Corresponding to our study, 10–20 wt.% PCL-grafted EC HAP (gHAP) remarkably improves the HAP dispersion in PMMA and enhances cell proliferation of osteoblast compared to nHAP nanoparticles. This suggests that cell proliferation properties of nano-composites are greatly influenced by the HAP dispersion.

## 3.5. Differentiation of MC3T3-E1 on the composite tablets

The ALP activity of cell differentiation on composite tablets is an indication of cell-to-cement interaction. The results are shown in figure 7. The ALP activity increased along with time in each group, and the results indicate that the ALP activity was improved by incorporating gHAP particles. However, the capacity of nHAP to improve the ALP activity was limited. In nHAP/PMMA composites, cellular ALP activity peaked at day 14 and decreased at day 21. In gHAP/PMMA composites, ALP activity increased until 21 days. On day 21, cells on gHAP/PMMA composites had significantly greater ALP activity than cells on nHAP/PMMA composites. The MC3T3-E1 cells on PMMAg20 composite had significantly higher ALP activity, even compared with PMMAg10. Again, the dispersion of nHAP in the PMMA matrix appears to be the main reason for this disparity in the ALP activity with the composite tablets. Homogeneous dispersion of gHAP in PMMA cement provided more effective interaction with the MC3T3-E1 cells. The homogeneity of the dispersion of nHAP appears to improve the functionality of the composites.

## 3.6. In vivo study

PMMAg20 was selected for the rabbit femur condyle defect model experiments because of the better in vitro biological activity. PMMAn20 was used for comparison. A critical size defect model was created and composite cement was implanted with a volume of approximately 0.1 ml (figure 8a,b).

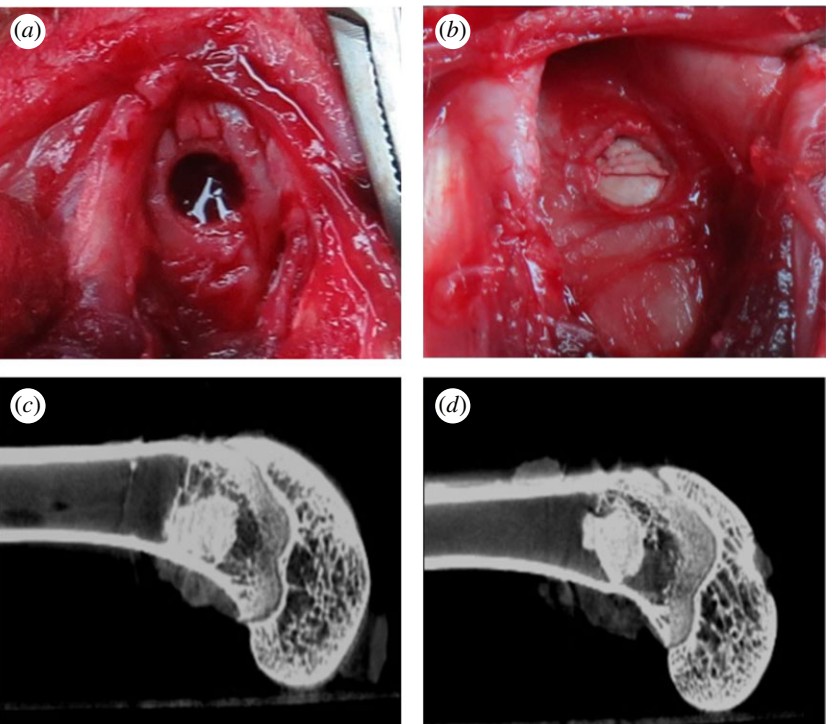

**Figure 8.** The image for PMMA composites implanted into bone tissue. (*a*) Appearance of the femoral condyle bone defect. The defect was implanted with injectable composites (*b*). Micro CT images of defect site were taken after three months' implantation with PMMAn20 (*c*) and PMMAg20 (*d*).

The micro CT images of the PMMAn20 and PMMAg20 were obtained after three months of implantation. The images showed that PMMA composite cement remained in the bone marrow cavities (figure 8*c*,*d*). In both groups, PMMA composites were incorporated in the osseous tissue without significant dislodgment.

## 3.7. Histology of PMMA-based composite cement *in vivo*

The *in vivo* biocompatibility of the PMMA-based cement was evaluated through histological H&E staining, as shown in figure 9. The PMMAn20 and PMMAg20 cements were observed to remain in the bone marrow cavities (marked with a white arrow) after implantation for three months, which corresponds with the micro CT results. It was observed that bone ingrowth capacity is better in PMMAg20 than in PMMAn20. The red box in figure 9*a* and *b* was magnified and is shown in figure 9*c* and *d*, respectively. The edge between the host bone and PMMA-based cements was pointed out with a yellow arrow as fibrous tissue (the dark purple area). Compared to PMMAn20, PMMAg20 had less fibrous tissue generation at its perimeter. Thus, PMMAg20 achieved a tight connection to the surrounding bone tissue. The homogeneous dispersion of gHAP is beneficial for bone ingrowth into the composite. This result agrees with previous cell viability, cell adhesion and the results of biological activity. The functionality of nHAP was hindered in PMMA-based cement due to poor dispersion. After modifying the surface of nHAP, the colloidal stability of gHAP was improved; moreover, the functionality of gHAP incorporated with PMMA-based cement was enhanced.

## 4. Conclusion

In this study, we have developed a PMMA-based composite cement with better mechanical strength and biocompatible properties by incorporating gHAP particles.

We harbour a simple and quick two-step method to graft PCL and EC to HAP, and produce the material gHAP. Evidently, the dispersion of HAP particles in PMMA is improved by using gHAP,

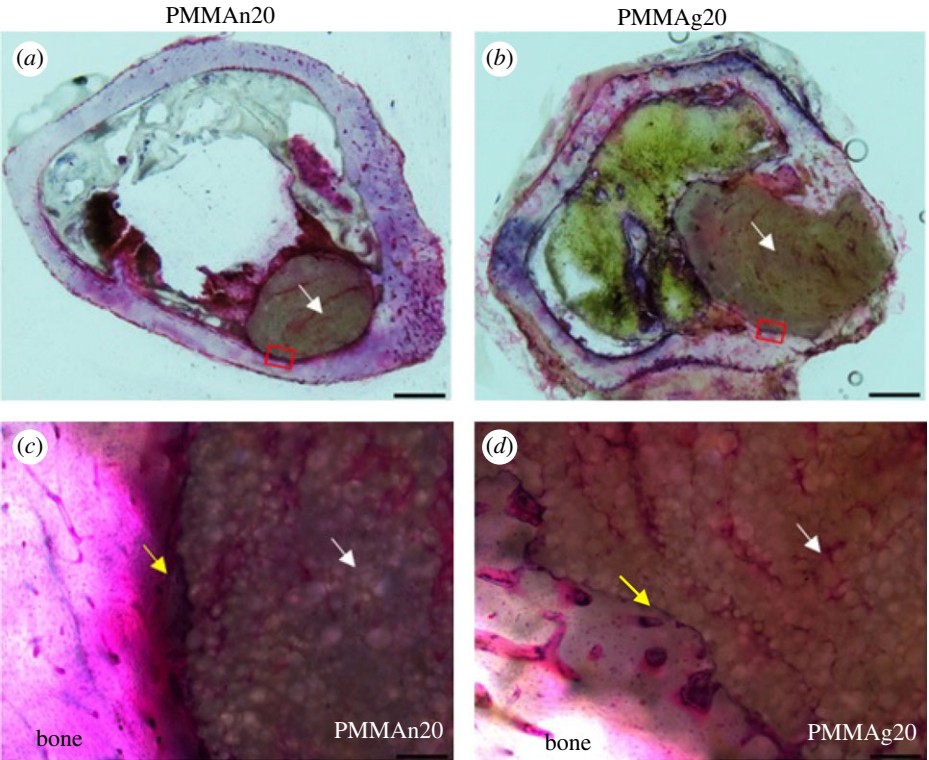

**Figure 9.** Histological images of bone tissue with PMMAn20 (*a*) and PMMAg20 (*b*) implanted after three months. The implanted cement was marked with a white arrow (scale bar 1000 μm). Magnified images of the red squares on PMMAn20 and PMMAg20 composite cements are shown in (*c*) and (*d*), respectively. PMMAn20 was separated from the host bone by a layer of fibrous tissue (yellow arrow); whereas, PMMAg20 had a tight connection to the surrounding bone tissue, which revealed better bone ingrowth (scale bar 20 μm).

and the homogeneous dispersion of gHAP increases the chance of particles interacting with cells. We have found that gHAP can maintain the mechanical properties of cement due to the better compatibility of the gHAP to the PMMA matrix. Because of homogeneous dispersion of gHAP, cell affinity and cell proliferation is promoted. Further, the fibrous tissue generated between cements and bone tissue is reduced after gHAP/PMMA implantation for three months. As a result, the PMMA-based cement with gHAP has better osteogenic activity and bone ingrowth. In conclusion, gHAP incorporation is a promising strategy for improving PMMA-based cements in many medical applications.

**Animal ethics.** The animal study was approved by the Animal Intuitional Review Board of Chung Gung Memorial Hospital (authorization number: 2014022101) and was conducted in compliance with the regulations for the care of laboratory animals.

**Data accessibility.** Data (electronic supplementary material) available from the Dryad Digital Repository at: http://datadryad.org/resource/doi:10.5061/dryad.bq18411.

**Authors' contributions.** Y.S.W. carried out the cellular laboratory work, participated in data analysis and drafted the manuscript; K.L.K. carried out the *in vivo* laboratory work, participated in the design of the study and drafted the manuscript; C.Y.W. participated in data analysis, carried out the statistical analyses, drafted the manuscript; D.W.H. conceived of the study, collect references and designed the study; Z.X.C. contributed to the analysis and interpretation of compressive strength data; C.A.H. contributed to analysis and interpretation of SEM and EDX data; I.M.C. and P.L.L. coordinated the study and helped draft the manuscript. All authors gave final approval for publication.

**Competing interests.** We declare we have no competing interests.

**Funding.** This study was supported by the grants from the Ministry of Science and Technology, Taiwan (NSC 106-2221-E-182-019-MY3) and Linkou Chang Gung Memorial Hospital Taoyuan, Taiwan (CRRPG3E0183 and CMRPG3H0211).

**Acknowledgements.** The authors appreciate the technical and equipment support from the Laboratory Animal Center in Linkou Chang Gung Memorial Hospital, Taiwan. The authors thank Zhi-Yi Chen for the technical support on the animal study.

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
