## [Reviewer comments · Royal Society Open Science]

Review History

RSOS-182060.R0 (Original submission)

Review form: Reviewer 1

Is the manuscript scientifically sound in its present form?

Yes

Are the interpretations and conclusions justified by the results?

No

Is the language acceptable?

Yes

Is it clear how to access all supporting data?

Yes

Do you have any ethical concerns with this paper?

No

Have you any concerns about statistical analyses in this paper?

No

Recommendation?

Accept with minor revision (please list in comments)

Comments to the Author(s)

1, At P17 ln23-41, the authors concluded that the cell distribution and proliferation was affected by the dispersion of HAP in PMMA. In Figure 2, the authors only showed the SEM images at low magnification. It is very hard for readers to distinguish the morphology differences of Figure 2A, D, E and F. Authors should also provide the enlarged SEM images to support the above conclusions.

2, The highlight of this manuscript is that the improved nHA dispersion can promote the cell affinity, proliferation, differentiation, and connection to the surrounding bone tissue. The manuscript mainly showed the experiment results. The underlying reasons should be deeply discussed with review of related literature.

Review form: Reviewer 2

Is the manuscript scientifically sound in its present form?

Yes

Are the interpretations and conclusions justified by the results?

Yes

Is the language acceptable?

Yes

Is it clear how to access all supporting data?

Not Applicable

Do you have any ethical concerns with this paper?

No

Have you any concerns about statistical analyses in this paper?

Yes

Recommendation?

Accept with minor revision (please list in comments)

Comments to the Author(s)

The manuscript has been well revised. However, the cell culturing results seemed not consistent as shown in Figure 4, 5 and 6. The authors are suggested to conduct cell culturing experiments again and give the results at day 1, 3, 5 and 7.

In addition, please explain the details of ANOVA method for comparison between each group as shown in Figure 3 and 7.

Decision letter (RSOS-182060.R0)

12-Mar-2019

Dear Dr Lai:

Title: Incorporation of Surface-modified Hydroxyapatite into Poly(methyl methacrylate) to Improve Biological Activity and Bone Ingrowth
Manuscript ID: RSOS-182060

Thank you for submitting the above manuscript to Royal Society Open Science. On behalf of the Editors and the Royal Society of Chemistry, I am pleased to inform you that your manuscript will be accepted for publication in Royal Society Open Science subject to minor revision in accordance with the referee suggestions. Please find the reviewers' comments at the end of this email.

The reviewers and handling editors have recommended publication, but also suggest some minor revisions to your manuscript. Therefore, I invite you to respond to the comments and revise your manuscript.

Because the schedule for publication is very tight, it is a condition of publication that you submit the revised version of your manuscript before 21-Mar-2019. Please note that the revision deadline will expire at 00.00am on this date. If you do not think you will be able to meet this date please let me know immediately.

Best wishes,
Dr Laura Smith
Publishing Editor, Journals

On behalf of the Subject Editor Professor Anthony Stace and the Associate Editor Professor Claire Carmalt.

RSC Associate Editor:
Comments to the Author:
(There are no comments.)

RSC Subject Editor:
Comments to the Author:
(There are no comments.)

Reviewer comments to Author:
Reviewer: 1

Comments to the Author(s)

1, At P17 ln23-41, the authors concluded that the cell distribution and proliferation was affected by the dispersion of HAP in PMMA. In Figure 2, the authors only showed the SEM images at low magnification. It is very hard for readers to distinguish the morphology differences of Figure 2A, D, E and F. Authors should also provide the enlarged SEM images to support the above conclusions.

2, The highlight of this manuscript is that the improved nHA dispersion can promote the cell affinity, proliferation, differentiation, and connection to the surrounding bone tissue. The manuscript mainly showed the experiment results. The underlying reasons should be deeply discussed with review of related literature.

Reviewer: 2

Comments to the Author(s)

The manuscript has been well revised. However, the cell culturing results seemed not consistent as shown in Figure 4, 5 and 6. The authors are suggested to conduct cell culturing experiments again and give the results at day 1, 3, 5 and 7.

In addition, please explain the details of ANOVA method for comparison between each group as shown in Figure 3 and 7.

Author's Response to Decision Letter for (RSOS-182060.R0)

See Appendix A.

Decision letter (RSOS-182060.R1)

05-Apr-2019

Dear Dr Lai:

Title: Incorporation of Surface-modified Hydroxyapatite into Poly(methyl methacrylate) to Improve Biological Activity and Bone Ingrowth
Manuscript ID: RSOS-182060.R1

It is a pleasure to accept your manuscript in its current form for publication in Royal Society Open Science. The chemistry content of Royal Society Open Science is published in collaboration with the Royal Society of Chemistry.

On behalf of the Subject Editor Professor Anthony Stace and the Associate Editor Professor Claire Carmalt.

RSC Associate Editor
Comments to the Author:
(There are no comments.)

Reviewer(s)' Comments to Author:

Appendix A

RESPONSES TO REVIEWERS'S COMMENTS

Thank you very much for your time and giving us chance to be considered for publication after minor revision. As shown below, we have addressed the reviewers' concerns, including the minor concerns and statistical analysis, point-by-point and have revised our original manuscript in keeping with reviewer points taken. Please let us know if there is anything else that we should address with respect to this manuscript.

Reviewer comments to Author:

Reviewer: 1

Comments to the Author(s)

1, At P17 ln23-41, the authors concluded that the cell distribution and proliferation was affected by the dispersion of HAP in PMMA. In Figure 2, the authors only showed the SEM images at low magnification. It is very hard for readers to distinguish the morphology differences of Figure 2A, D, E and F. Authors should also provide the enlarged SEM images to support the above conclusions.

RESPONSE:

We agree with you that it is hard for reader to distinguish the differences of Figure 2A, D, E and F. We have replaced with new SEM images to make the point clearer. SEM was used to observe HAP, which is 200 nm. If HAP aggregates together, the size will become larger and present as bright spots on the SEM image. Thus, some bright spots were significantly observed on the surface of PMMA_n10 and PMMA_n20 tablets (Figure 2A,B). The bright spot on the surface of PMMA_n20 were confirmed for the composition containing calcium and phosphate using EDX analysis (Figure 2C). Conversely, there were fewer bright spots observed on the surfaces of PMMA_g10 and PMMA_g20 (Figure 2D,E), which indicates that no significant particle agglomeration occurred in these two tablets. Plain PMMA surface are also presented for comparison and had smooth surfaces with some small dots.

2, The highlight of this manuscript is that the improved nHAP dispersion can promote the cell affinity, proliferation, differentiation, and connection to the surrounding bone tissue. The manuscript mainly showed the experiment results. The underlying reasons should be deeply discussed with review of related literature.

RESPONSE:

Really appreciated for your kindly suggestion. We have added some discussion about the correlation between HAP dispersion and cell growth as below and this discussion is also added in the final paragraph on the page 17-18.

“Kim et al. have reported that cell proliferation of mouse fibroblast is significantly enhanced on 24 wt.% PCL grafted HAP nano-composites compared to the PCL only because of excellent dispersion [31]. Keivani et al. have demonstrated that even addition of 7 wt.% PCL grafted HAP nanoparticles to PCL is enough to significantly increase cell proliferation of fibroblast as well [32]. Corresponding to our study, 10-20 wt.% PCL-grafted EC HAP (gHAP) remarkably improves the HAP dispersion in PMMA and

enhances cell proliferation of osteoblast compared to nHAP nanoparticles. Thus, these literatures suggest that cell proliferation property of nano-composites is greatly influenced by the HAP dispersion.”

Reviewer: 2

Comments to the Author(s)

1) The manuscript has been well revised. However, the cell culturing results seemed not consistent as shown in Figure 4, 5 and 6. The authors are suggested to conduct cell culturing experiments again and give the results at day 1, 3, 5 and 7.

RESPONSE:

Thank you very much for your suggestion. The previous descriptions maybe not clear enough. We have pointed out the different experimental condition of Figure 4, 5 and 6.

- (a) In Figure 4 and 5, cells were seeded on the composites for 1 day and then assayed by Live & Dead staining and CCK-8, respectively. These results indicate the cell attachment on the composites. In Figure 6, we cultured cells on the composites for 3 days and then observed cell morphology and cell number by SEM.
- (b) The magnification of image is higher in Figure 6 compare to Figure 4.

The data show that cell attachment is slightly different between nHAP/PMMA and gHAP/PMMA groups but lower in PMMA only (Figure 4, 5); however, after 3-day incubation, cell proliferation is significantly upregulated on gHAP/PMMA composite with better dispersion, but not on nHAP/PMMA and PMMA only. From these data, cell culturing for 3 day on composites is enough to observe the difference of cell proliferation between gHAP/PMMA and nHAP/PMMA.

2) In addition, please explain the details of ANOVA method for comparison between each group as shown in Figure 3 and 7.

RESPONSE:

Thank you very much for kindly pointing out this typo. In Figure 3 and 7, we use unpaired two-tailed Student's *t* test to compare two groups. We have modified the description in “Material and Method→ Statistical Analysis” on the page 11:

“Statistical analysis and the results are shown as the mean ± SD. Significant differences were evaluated by using unpaired two-tailed Student's *t* test. The level of statistically significant was set at $p < 0.05$. *, $p < 0.05$; **, $p < 0.01$; ***, $p < 0.001$.”